# Open-ended Commonsense Reasoning with Unrestricted Answer Scope

**Chen Ling[1], Xuchao Zhang[2], Xujiang Zhao[3], Yanchi Liu[3], Wei Cheng[3],**
**Mika Oishi[4], Takao Osaki[4], Katsushi Matsuda[4], Haifeng Chen[3], Liang Zhao[1]**
[1]Emory University, [2]Microsoft, [3]NEC Labs America, [4]NEC Corporation
chen.ling@emory.edu, xuzhao@nec-labs.com

## Abstract

Open-ended Commonsense Reasoning is defined as solving a commonsense question without providing 1) a short list of answer candidates and 2) a pre-defined answer scope. Conventional ways of formulating the commonsense question into a question-answering form or utilizing external knowledge to learn retrieval-based methods are less applicable in the open-ended setting due to an inherent challenge. Without pre-defining an answer scope or a few candidates, open-ended commonsense reasoning entails predicting answers by searching over an extremely large searching space. Moreover, most questions require implicit multi-hop reasoning, which presents even more challenges to our problem. In this work, we leverage pre-trained language models to iteratively retrieve reasoning paths on the external knowledge base, which does not require task-specific supervision. The reasoning paths can help to identify the most precise answer to the commonsense question. We conduct experiments on two commonsense benchmark datasets. Compared to other approaches, our proposed method achieves better performance both quantitatively and qualitatively. Our code and data are available at: https://github.com/lingchen0331/KEEP.

## 1 Introduction

Current research on commonsense reasoning conventionally formulates the problem into a multiple-choice question answering (QA) format, where the best answer is expected to be chosen from a list of candidates for the given commonsense question. However, there are many practical and real-world scenarios where a small list of answer candidates or an answer scope (i.e., a relatively large set of concepts where the correct answer exists) are missing or not even provided (e.g., arbitrary questions asked in the search engine), which requires the intelligent system to *understand* the commonsense question rather than picking a correct answer from

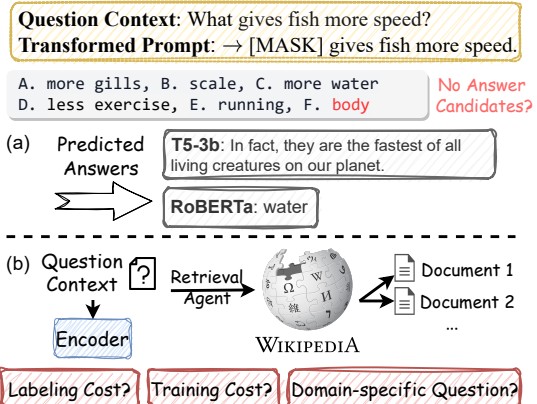

Figure 1: Current approaches can only partially solve the open-ended commonsense reasoning with unrestricted answer scope.

a pre-defined pool. In this study, we focus on the **open-ended commonsense reasoning**, where we answer commonsense questions with two **constraints**: i.e., *without regulating an answer scope* and *without a pre-defined answer candidates list*. Open-ended commonsense reasoning is inherently challenging due to its core obstacle: the unrestricted answer scope would result in an extremely large searching space, where the model cannot retrieve relevant answers effectively and efficiently.

Two types of approaches can be adapted to partially solve the open-ended commonsense reasoning (i.e., solving only one constraint). On the one hand, PLMs have been demonstrated to excel in various NLP tasks by using prompts. However, as shown in Figure 1 (a), both RoBERTa-large (Liu et al., 2019) and T5-3b (Kale and Rastogi, 2020) may not provide satisfying answers to the commonsense question since they can only leverage their own corpus to fill the mask in prompts. Without providing answer candidates, PLMs may have a limited capacity to obtain and accurately predict the answer that requires structured reasoning. Even though lots of methods (Lin et al., 2019; Yasunaga et al., 2021; Zhang et al., 2021) have emerged to incorporate external knowledge bases and PLMs to perform

joint reasoning, their methods still require a small set of answer candidates, which is not applicable in the open-ended scenario.

To resolve the necessity of a few answer candidates in the open-ended QA problem, researchers have developed various knowledge-augmented retrieval methods, and their general inference scheme is visualized in Figure 1 (b). Specifically, instead of regulating a small set of answer candidates, knowledge-augmented retrieval methods (Ma et al., 2021; Bian et al., 2021; Lin et al., 2021) have designed an *answer scope* that directly contains the correct answer, and they leverage learning-based ranking algorithms to select the best answer. Although the form of the answer scope can vary, including a large set of conceptual entities and a set of question-related documents, building such an answer scope is still a resource-consuming and *ad-hoc* process. Moreover, well-trained retrievers are dependent on specific answer scopes, which are less applicable in real-world applications. For example, it's impossible to provide a relevant document set when answering commonsense questions during a conversation with a chatbot.

In this work, we present the external K̲nowlE̲dge-E̲nhanced P̲rompting method (KEEP) to achieve open-ended commonsense reasoning without pre-defining an answer candidate set and an answer scope. Firstly, to eliminate the requirement of answer candidates, KEEP leverages an external knowledge base (e.g., ConceptNet) as the answer searching space and iteratively extracts multi-hop reasoning paths relevant to the question. To avoid searching exhaustively over the whole knowledge base, we leverage PLMs to formulate the overall search criteria. The key insight is PLMs have certain reasoning abilities through their large-scale model parameters, which can be utilized to provide implicit knowledge in determining whether or not to keep expanding the reasoning paths or adopt the entity in the path as the final answer. Therefore, without restricting specific answer scopes and direct supervision of the reasoning process, KEEP can be applied in most real-world scenarios requiring commonsense reasoning. To further enhance the reasoning ability of the PLM, we propose to leverage task-agnostic reasoning paths extracted directly from the external knowledge base as training instances to finetune the PLM.

We summarize our main contributions as follows. *a*) We formulate the open-ended commonsense reasoning problem as a multi-hop reasoning task iteratively conducted on an external knowledge graph. *b*) We leverage the implicit knowledge stored in PLMs to guide the overall searching/reasoning process under both zero-shot and finetuning settings. *c*) We utilize the retrieved reasoning paths as additional explanations to justify the answer choice. *d*) We empirically demonstrate the performance of our method against the state-of-the-arts, which excels other comparison methods in multiple metrics under the open-ended setting.

## 2 Related works

**Neural Commonsense Reasoning.** Combining PLMs and external knowledge for reasoning has recently gained lots of attention (Chen et al., 2020; Chowdhury et al., 2023). State-of-the-art methods have been invented to inject commonsense knowledge into language models, either by pre-training on knowledge bases (Ma et al., 2021; Chang et al., 2021), finetuning the model on the test domain (Bian et al., 2021), or leveraging structured knowledge base (e.g., ConceptNet) (Yasunaga et al., 2021; Zhang et al., 2021; Cui et al., 2023) so that they can infer with additional retrieved knowledge. However, none of these works except for LLMs (Ling et al., 2023c) can be trivially adapted to solve the open-ended commonsense reasoning since they require substantial training instances for pre-training/finetuning or a list of pre-existing answer candidates designed for the question.

**Open-ended Commonsense Reasoning.** To date, a few attempts are trying to solve the open-ended commonsense reasoning. Gerber et al. (2010) and Roemmele et al. (2011) have first formulated the open-ended commonsense reasoning problem and leveraged the statistical natural language processing techniques. However, their performance is rather limited, and they may only provide a series of knowledge statements instead of providing a plausible answer. With the development of deep language models, a number of conversational and Mask Language Models (MLMs) (e.g., GPT-3 (Brown et al., 2020) and RoBERTa (Liu et al., 2019)) can also perform the task by leveraging prompt tuning and learning. However, even the powerful GPT-3 may not provide satisfying answers in the zero-shot setting by only relying on its own corpus. In addition, our work is also related to the work of *open-ended* commonsense reasoning (Lin et al., 2021), which formulated open-ended commonsense reasoning

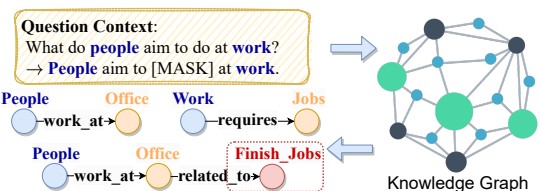

Figure 2: Example of the open-ended commonsense reasoning: the model takes the question as input and returns supporting reasoning paths (i.e., knowledge statements) with the best answer.

as a concept ranking process. Their approach still entails a training procedure on a given document set that is related to the commonsense question, which deviates from the main purpose of open-ended commonsense reasoning: lack of pre-defined answer candidates and finetuning data.

**Explanation Generation for Commonsense Reasoning.** Other than predicting the correct answer, it is also important to explore explicit reasoning steps behind the answer selection. Other than works that require direct supervision to predict explanation (Paranjape et al., 2021), Bosselut et al. (2021) proposed to leverage knowledge graphs to acquire reasoning paths as the explanation in an unsupervised way. However, this approach requires pre-defined answers to guide the reasoning, which is not applicable in open-ended commonsense reasoning. The other line of works (Shwartz et al., 2020; Ji et al., 2020; Liu et al., 2021; Ling et al., 2023b) have also been utilizing model-generated text as the clarification of the commonsense question and empirically demonstrating the performance can be boosted by augmenting the query with knowledge statements. However, their models still require answer candidates as the model input. Additionally, purely relying on the language model still lacks the model transparency, and the generated knowledge statement cannot be empirically served as the answer explanation (Liu et al., 2021).

## 3 Proposed Method

In this section, we first introduce the problem formulation, and then discuss the detailed framework of the proposed method, which can be divided into three components: 1) entity extraction and linking, 2) local knowledge graph expansion, and 3) training strategy and answer prediction.

### 3.1 Problem Formulation

We aim to solve open-ended commonsense reasoning questions by jointly using knowledge from a *PLM* and a *structured knowledge graph G*. The

knowledge graph (KG) $G = (V, E)$ (e.g., Concept-Net) is a multi-relational heterogeneous graph (Ling et al., 2021, 2023a). $V$ is the set of entity nodes, $E \subseteq V \times R \times V$ is the set of edges that connect nodes in $V$, where $R$ represents a set of relation types (e.g., *locates_at* or *requires*). Specifically, given an open-ended commonsense reasoning question $q$ **without** *providing answer candidates* and *regulating an answer scope*, the target of this work is to determine 1) a local KG $G_q \in G$ contains relevant information of $q$; 2) a set of reasoning paths $\mathbf{k} = \{k_1, k_2, ..., k_m\}$ extracted from $G_q$; and 3) an entity $\hat{a}$ extracted from $\mathbf{k}$ that is precise to answer the question $q$. For example, in Figure 2, to answer a commonsense question *"what do people aim to do at work?"*, we aim at first extracting all relevant reasoning paths from the external KG that can provide us with logical information to answer the question. Among all the paths, we select the most precise one (i.e., *people → office → finish_jobs*) and extract the answer $\hat{a} = finish\_jobs$ such that the following joint likelihood can be maximized.

$$P(\hat{a}, \mathbf{k}|q, G_q) = P(\mathbf{k}|q, G_q) \cdot P(\hat{a}|\mathbf{k}) \quad (1)$$

**Challenges.** However, maximizing the joint likelihood in Equation (1) is not a trivial task due to two critical obstacles. First, retrieving the question-relevant reasoning paths $\mathbf{k}$ (i.e., knowledge statements) is difficult since we cannot build a local KG between question entities and answer candidates under the open-ended setting as Yasunaga et al. (2021); Zhang et al. (2021); Lin et al. (2019); Lu et al. (2023) do. Moreover, without regulating a pre-defined answer scope as Lin et al. (2021) does, the search space would be the whole knowledge graph. Exhaustively expanding a multi-hop neighborhood that is relevant to the question on the knowledge graph would cause severe scalability issues.

Next, to solve both challenges, we discuss how to initiate the local KG and iteratively reason over it to find all plausible knowledge statements and the most convincing answer. We demonstrate the overall framework in Figure 3.

### 3.2 Local Graph Construction and Expansion

**Knowledge Graph Entity Linking.** Conceptual knowledge graphs (e.g., ConceptNet) enable a variety of useful context-oriented reasoning tasks over real-world texts, which provides us with the most suitable structured knowledge in open-ended commonsense reasoning. To reason over a given commonsense context using knowledge from both PLM

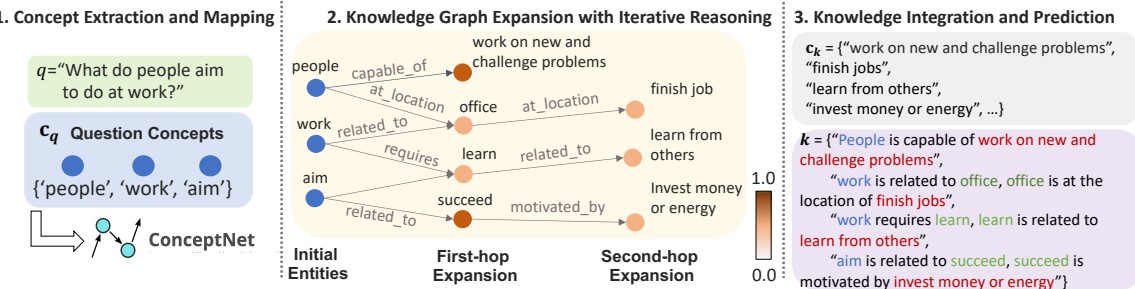

Figure 3: The framework of the proposed method consists of 1) concept extraction and entity linking; 2) local knowledge graph expansion with iterative reasoning steps, and 3) knowledge integration and final answer prediction.

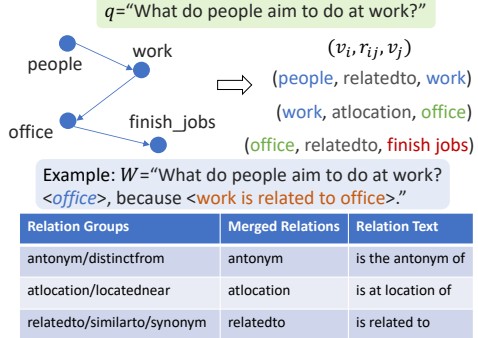

Figure 4: Knowledge statement transformation and cloze-based prompt construction.

and $G$, the first step of the framework is to extract the set of critical entities $\mathbf{c}_q = \{c_q^{(1)}, ..., c_q^{(i)}, ...\}$ from the question $q$ that have the surjective mapping to a node set $V_q \in V$ in the KG. Since $q$ is often presented in the form of non-canonicalized text and contains fixed phrases, we follow the prior work (Becker et al., 2021) to map informative entities $\mathbf{c}_q$ from $q$ to conjunct concept entities $V_q$ in KG by leveraging the latent representation of the query context and relational information stored in $G$.

**Reasoning Over Local Knowledge Graph.** To imitate the human reasoning process, we aim to retrieve reasoning paths within $L$ hops from $G$ to form the local knowledge subgraph $G_q$ that has the highest coverage to the question concepts $\mathbf{c}_q$. Ideally, each path in $G_q$ can be regarded as a reasoning chain that helps to locate the most precise answer and its explanation to the question $q$. However, expanding $L$-hop subgraph $G_q$ from $\mathbf{c}_q$ is computationally prohibited. Unlike other works (Yasunaga et al., 2021; Lin et al., 2019) that build $G_q$ between the question $q$ and all answer candidates, the open-ended commonsense reasoning problem does not provide any directions (i.e., answer candidates) or limit the answer scope. The typical node size of a 3-hop local KG with $|\mathbf{c}_q| = 3$ could easily reach $1,000$ on ConceptNet, and many nodes are irrelevant under the current question context.

**Reasoning Path Pruning.** In order to make the process of reasoning path expansion scalable, we incorporate the implicit knowledge in PLMs to prune irreverent paths. Specifically, we pair the question $q$ with the text of node $v$ along with the reasoning-path-transformed knowledge statement to form a cloze-based prompt $W = [q; v_j; (v_i, r_{ij}, v_j)]$ in order to turn the local graph expansion problem into an explicit reasoning procedure by directly answering the question with its derived reasoning path. For example, in Figure 4, the prompt is formatted as *What do people aim to do at work? <answer_node>, because <reasoning path>*. We leverage a pre-defined template to transform the triplet $(v_i, r_{ij}, v_j)$ into natural language. For instance, the triplet (*work, antonym, unemployment*) can be translated to *work is the antonym of unemployment* as illustrated in Figure 4. Note that a KG typically contains many edge types that have similar meanings (e.g., both *antonym* and *distinct_from* have the same meaning *antonym*); therefore, we merge similar edge types into a unified template and illustrate a few examples of the templates in Figure 4. To evaluate whether we keep the reasoning path, we propose leveraging the PLM to score the relevance of each reasoning path given the context of the question. Formally, suppose the prompt $W$ consists of $N$ tokens $W = \{\omega_1, ..., \omega_{n-1}, \omega_n, \omega_{n+1}, ..., \omega_N\}$, the commonsense score $\phi_l(W)$ of the logical sentence $W$ composed at $l$-th hop expansion is defined as:

$$\phi_l(W) := \sum_{n=1}^{N} \log(p_\theta(\omega_n | W_{\setminus n}))/N, \quad (2)$$

where the $W_{\setminus n}$ indicates replacing the token $\omega_n$ to the [MASK], and the denominator $N$ reduces the influence of the sentence length on the score prediction. Intuitively, $\log(p_\theta(\omega_n | W_{\setminus n}))$ can be interpreted as how probable a word $\omega_n$ given the context. For example, by filling *blue* and *red* into the masked logical statement $W_{\setminus n}$ = *The sky is* [MASK], *blue* should have a higher score than *red*.

As we iteratively expand $G_q$, each $\phi_l(W)$ scores a unique reasoning path at a particular $l \in [1, L]$

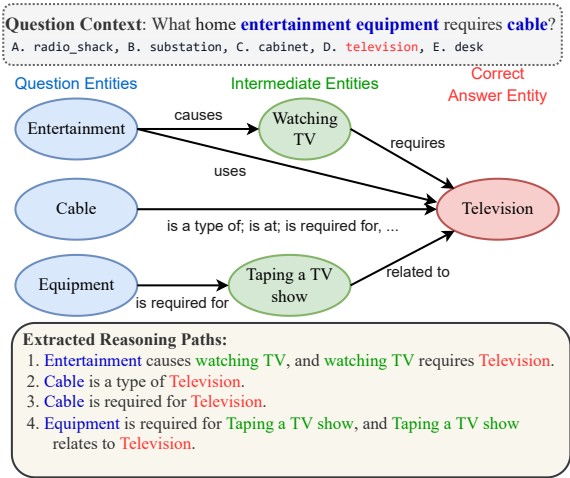

**Question Context**: What home **entertainment equipment** requires **cable**?
A. radio_shack, B. substation, C. cabinet, D. television, E. desk

**Extracted Reasoning Paths:**
1. Entertainment causes watching TV, and watching TV requires Television.
2. Cable is a type of Television.
3. Cable is required for Television.
4. Equipment is required for Taping a TV show, and Taping a TV show relates to Television.

Figure 5: Training Corpus Generation. For each commonsense question in the training set, we discover all the reasoning paths between entities in the question and the correct answer entity in ConceptNet. All the reasoning paths are transformed into sentences by templates and thus serve as the finetuning corpus of our model.

depth in the graph. As marked in Figure 3, a higher score $\phi_l(W)$ indicates the node $v_j$ should be kept for the next $(l + 1)$ hop expansion.

### 3.3 Training Strategy and Answer Prediction

**Training Strategy.** The proposed framework is able to answer open-ended commonsense questions with any off-the-shelf language models and the ConceptNet. However, we empirically find the performance of the off-the-shelf PLM is rather limited (i.e., commonsense score $\phi_l(\cdot)$ is less distinguishable between different prompts) when dealing with long-range reasoning paths (e.g., $L \geq 2$). In order to further enhance the PLM's reasoning capability, we propose to finetune PLMs on the knowledge examples constructed from ConceptNet. Specifically, we aim to enhance the $p_\theta$'s reasoning capability by correctly identifying the knowledge triplets on ConceptNet. As depicted in Figure 5, given a commonsense question $q$ = *"What home entertainment equipment requires cable?"* and its correct answer $\tilde{a}$ = *"television"*, we identify reasoning paths $[(v_1, r_1, v_2), ..., (v_{L-1}, r_{L-1}, v_L)]$ on $G$ from each entity $c_q^{(i)}$ in $\mathbf{c}_q$ to $\tilde{a}$. Note that there may exist multiple paths $c_q^{(i)}$ to $\tilde{a}$; e.g., *"Cable is a type of Television"* and *"Cable is required for Television"*. Each reasoning path is then transformed as natural language sentences with templates as illustrated in the table of Figure 4. We follow the standard masked language modeling task to finetune the model. By randomly masking a small portion (i.e., 15%) of tokens in each sentence, we aim to let

the PLM comprehend the latent logic behind each retrieved reasoning path by learning to fill masks.
**Answer Prediction.** After we obtained the subgraph $G_q$ consisting of all reasoning paths $\mathbf{k}$ within $L$-hop with a high commonsense score, each path $k_i \in \mathbf{k}$ can be regarded as an individual supporting knowledge explanation to an answer $a_i$.

$$\log p_\theta(a_i|k_i) \propto \phi_L = \sum_{l=1}^{L} \phi_l,$$

where the $\phi_L$ denotes the final score for each answer $a_i$ within $L$-hop and can be interpreted as approximating the likelihood of answer $a_i$ given a singular reasoning path $\{c \rightarrow v_1 \rightarrow \cdots \rightarrow a\}$. To better improve efficiency, we utilize beam search to only keep high-confidence reasoning paths. We can thus pick the answer $\hat{a}$ and its reasoning path $\hat{k}$ with the highest score $\phi_L$ as the final answer and supporting knowledge.

## 4 Experiment

We leverage RoBERTa-large (Liu et al., 2019) as our base PLM. We empirically verify the performance of the proposed method against other methods on commonsense reasoning benchmark datasets under the open-ended setting. Due to the space limit, more experiments, case studies, and implementation details can be found in Appendix A.1.

### 4.1 Experiment Setting

**Dataset.** We evaluate our method on two commonsense reasoning benchmarks. *1) CommonsenseQA (CSQA)*: Talmor et al. (2019) that contains 1, 140 test cases. *2) QASC*: Khot et al. (2020) that contains 917 test cases. We only keep the question and discard the attached multiple-choice answers.
**Comparison Methods.** Since we are the first to investigate open-ended commonsense reasoning with unrestricted answer scope, we have no direct opponents to compete. All the QA models either require pre-defined answer candidates or a specific answer scope, which *CANNOT* be applied in the real open-ended scenario. In this work, we compare our model against the following baselines. 1) *RoBERTa-large* (Liu et al., 2019) is a MLM that is trained with dynamic masking, a larger batch size, and a larger vocabulary size. 2) *DeBERTa-v3-large* (He et al., 2021) improves the BERT and RoBERTa models using disentangled attention and enhanced mask decoder. 3) *RelBERT* (Ushio et al., 2021) is a finetuned model based on RoBERTa, which

| Method | | CSQA | | | QASC | | |
|---|---|---|---|---|---|---|---|
| | | Top-1 | Top-3 | Top-5 | Top-1 | Top-3 | Top-5 |
| Masked Language Model | DeBERTa-v3-large | 0.273 | 0.426 | 0.607 | 0.254 | 0.554 | 0.618 |
| | RoBERTa-large | 0.275 | 0.477 | 0.682 | 0.294 | 0.523 | 0.578 |
| | RelBERT | 0.302 | 0.567 | 0.698 | 0.362 | 0.574 | 0.601 |
| Generative Language Model | T5-3b | 0.426 | 0.471 | 0.501 | 0.422 | 0.546 | 0.572 |
| | UnifiedQA | 0.395 | 0.439 | 0.517 | 0.379 | 0.513 | 0.602 |
| | GPT-3 | 0.476 | 0.654 | 0.769 | 0.452 | 0.573 | 0.749 |
| Ours | KEEP (w/o finetuning) | 0.385 | 0.615 | 0.776 | 0.467 | **0.742** | 0.821 |
| | KEEP | **0.523** | **0.714** | **0.798** | **0.489** | 0.732 | **0.829** |

Table 1: Top-1, 3, and 5 prediction accuracy made by human annotators for each model. (The higher the better)

particularly focuses on improving the relation embedding and leverages relational triplets extracted from ConceptNet as training corpus. 4) *T5-3b* (Kale and Rastogi, 2020) is an encoder-decoder-based language model pre-trained on a multi-task mixture of unsupervised and supervised tasks. We use its 3b version that contains 3 billion parameters. 5) *UnifiedQA* (Khashabi et al., 2020) is a unified pre-trained language model specifically for generative question-answering tasks, which is based on T5-large. 6) *GPT-3* (Brown et al., 2020) is one of the largest language models with 175 billion parameters, which is powerful and excels other methods in multiple NLP tasks. We use all language models in the zero-shot setting to follow the open-ended application scenario.

**Evaluation Criteria.** Since we do not have ground truth to evaluate the prediction correctness, we generate answer candidates for each commonsense question and work with human annotators to indicate whether there exists a precise answer that could answer the given question. For baselines with MLMs: RoBERTa, DeBERTa, and RelBERT, we design prompts that allow each model to fill the mask with top-$N$ answer choices. For baselines with generative language models, T5-3b, UnifiedQA, and GPT-3, they are prompted to generate direct answers (within 20 tokens). In addition to human judgment, we also incorporate the commonsense score (Equation (2)) to evaluate the perplexity of the answer choice. Specifically, we choose the best answer from all the candidates generated by each model and concatenate the answer to the original question. Following the way of evaluating the commonsense score of the sentence in Zhou et al. (2020), We use GPT2-large (Radford et al., 2019) and LLaMA2-70B (Touvron et al., 2023) as the base model to calculate the score since GPT2-large is not included in our comparison methods. Intuitively, a PLM should assign higher probabilities to answers that are semantically and syntactically

| Method | GPT-2 | | LLaMA-2-70B | |
|---|---|---|---|---|
| | CSQA | QASC | CSQA | QASC |
| DeBERTa-large | 12.498 | 15.528 | 94.391 | 62.497 |
| RoBERTa-large | 8.589 | 10.788 | 91.467 | 59.582 |
| RelBERT | 9.327 | 8.543 | 77.813 | 54.284 |
| T5-3b | 9.152 | 9.314 | 76.789 | 56.759 |
| UnifiedQA | 8.573 | 8.439 | 58.929 | 45.621 |
| GPT-3 | 6.527 | **7.528** | 43.325 | 33.569 |
| KEEP | **6.139** | 7.692 | 47.472 | 37.793 |
| Ground Truth | 4.844 | 5.327 | 34.675 | 29.579 |

Table 2: The commonsense score of each model, which is calculated by GPT-2 and LLaMA-2 through concatenating each question and the most suitable answer generated from each model. (The lower the better)

correct to the question.

### 4.2 Results

**Qualitative Analysis.** Table 1 summarizes the Top-*N* accuracy results. For each approach, the test results are obtained by evaluating if there is a precise answer in the Top-1, 3, and 5 generated answers. As shown in the table, our proposed method excels both MLMs and generative language models by an evident margin (achieved approximately 9%, 20%, and 15% improvement than the second-best on Top-1, 3, and 5 accuracy on both datasets, respectively). Additionally, We report several observations from the table to explain the results: *1) PLMs do not generalize well on unseen entities.* Without relying on pre-defined answer candidates, PLMs do not make satisfied predictions of reasoning-related prompts. Especially for MLMs like DeBERTa and RoBERTa, most of the correct answers in the reasoning questions are not even encountered during pre-training due to their heavy reliance on memorization in the pre-training process. *2) PLMs are becoming a promising alternative to external knowledge bases.* As we can see from the table, generative PLMs (i.e., T5, UnifiedQA, and GPT-3) generally perform well on the Top-1 accuracy on both datasets, which indicates signs of capturing relational knowledge in a zero-shot setting reasonably well compared to our

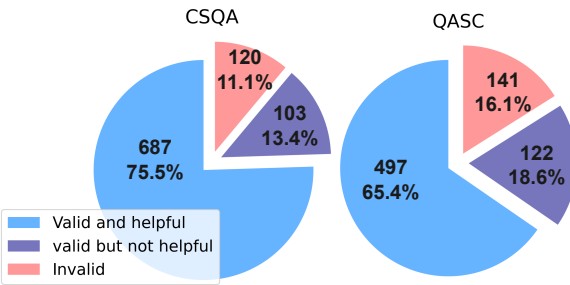

**Example of Invalid Reasoning Path (Links wrong entities from question to the correct answer):**
**Question**: Where is a business restaurant likely to be located?
**Answer**: Big City
**Generated Reasoning Path**: Business is located at big city.

**Example of Valid but Not Helpful Reasoning Path (Links correct entities from the question but not helpful):**
**Question**: Johnny sat on a bench and relaxed after doing a lot of work on his hobby. Where is he?
**Answer**: Terrace
**Generated Reasoning Path**: The synonym of the bench is terrace.

Figure 6: The Percentage of Valid Reasoning Paths in the Correct Prediction. There are 910 correct predictions in the CSQA dataset, and 809 of them are valid. In QASC dataset, there are 638 out of 760 reasoning paths that are valid based on human judgment.

| Setting | Top-1 Score | |
|---|---|---|
| | CSQA | QASC |
| Ours (Reasoning Path Length $L = 3$) | 0.523 | 0.489 |
| w/o finetuning | 0.385 | 0.467 |
| w/o explanation | 0.449 | 0.423 |
| Reasoning Path Length $L = 1$ | 0.379 | 0.422 |
| Reasoning Path Length $L = 2$ | 0.515 | 0.476 |

Table 3: Ablation Study. We report the performance of KEEP under different settings.

proposed method. However, their performances do not show evident improvements if we can choose from Top-3 and 5 candidates. Even though MLMs can achieve improvements at each level, their performance still cannot be compared to ours since standard PLMs lack knowledge awareness without accessing external knowledge. *3) Commonsense reasoning ability is not fully determined by the model size.* Without proper finetuning on target datasets, larger language models may find it harder to mine latent and unstructured knowledge, which indicates their performance may deteriorate. For instance, RoBERTa-large and RelBERT contain 300 million of parameters while T5-3b contains more than 3 billion of parameters. However, both RoBERTa-large and RelBERT have competitive performance with T5-3b on both datasets.

**Quantitative Analysis.** Table 2 depicts the average commonsense score of the predicted answer from each comparison method. Specifically, our human annotators pick the most suitable answer from each

model's predicted answer candidates, and we concatenate the answer with the question to form a sentence. We use the vanilla GPT-2-large to calculate the commonsense score of each sentence by Eq. (2). As shown in Table 2, our method achieves competitive performance with GPT-3 across two datasets. First, lacking support from external knowledge bases and pre-defined answer candidates, MLMs may not perform well in generating answers only relying on prompts. Generative models tend to generate long and coherent sentences as the answer rather than short words/phrases. Even though they may not fit the commonsense, they can still achieve better commonsense scores than MLMs.

**Validity of Reasoning Paths.** In addition, we also incorporate human evaluation to check the validity of the generated reasoning paths for our methods, and we report the percentage of valid reasoning paths in Figure 6. Note that the invalid reasoning path is defined as falsely linking the correct entity in the question to the semantically correct answer, and the valid but not helpful reasoning paths denote our model links the correct entity from the question to a semantically correct answer on ConceptNet, but the reasoning path may not aid the language model make predictions. We also give examples of both cases in the figure. Statistics show that the majority (nearly 90% on the CSQA and 80% on the QASC) of the generated reasoning paths are grammatical and valid to the question. On top of that, around 70% of them can be helpful and relevant to the context of the question.

**Ablation Study.** Next, we conduct an ablation study to investigate the importance of each component in the model, and the results are reported in Table 3. Firstly, the performance of our method drops around 25% without finetuning, which indicates the logical sentences transformed from reasoning paths can indeed help the model navigate to the most correct answers that fit the commonsense in ConceptNet. Secondly, we discard concatenating the reasoning-path-transformed explanation to the prompt, and the performance also drops approximately 15% in both datasets. As with other explanation-aided commonsense reasoning models (Shwartz et al., 2020; Liu et al., 2021), the logical sentence can indeed help the model make better predictions. Finally, we also investigate how the length of reasoning paths impacts the model performance. Specifically, the length of reasoning paths denotes the maximal hop of neighbors our method could explore from

| CSQA Test Case | Generation Results |
|---|---|
| | **Prompt**: What do people aim to do at work?
→(People aim to [MASK] at work.) |
| DeBERTa-v3-large | burst |
| RoBERTa-large | succeed |
| RelBERT | improve |
| T5-3b | What tools are they going to use? What products is their life like? |
| UnifiedQA | The answer is not to just go. "Work" is only part of the solution to unemployment. |
| GPT-3 | People aim to do many different things, depending on their individual goals and aspirations. |
| KEEP (Ours) | Work on new and challenging problems.
**Reasoning Chain**:
**Work** is done by **People**, **People** desires to **work on new and challenging problems**. |
| QASC Test Case | Generation Results |
| | **Prompt**: What is saturated fat at room temperate?
→(The saturated fat at room temperate is [MASK].) |
| DeBERTa-v3-large | unchanged |
| RoBERTa-large | negligible |
| RelBERT | zero |
| T5-3b | It is a major source of energy in the human body. |
| UnifiedQA | You will find fats like *butter* or *margarine*, the main components of the food chain. |
| GPT-3 | Saturated fat is a type of fat that is solid at room temperature. |
| KEEP (Ours) | Solid Object.
**Reasoning Chain**:
**Fat** is related to **Butter**, **Butter** is a type of **solid object**. |

Table 4: Test cases of all comparison methods on both datasets. Under the open-ended setting, KEEP excels in other methods and achieves competitive performance with GPT-3 in generating answers and valid reasoning paths.

the question entities. Intuitively, if we regulate the length of reasoning paths to be short, it may not reach answers that require multi-hop reasoning. However, if we set a large path length, the model may generate noisy paths and the search time would be unacceptable. As shown in the table, there is a large performance gap if we set the reasoning path length to be 1, which indicates most of the answers do not exist within the first hop of question entities. Ae increase the length, the performance difference between $L = 2$ and $L = 3$ (our adopted length) is very small. Therefore, considering both effectiveness and efficiency, we adopt $L = 3$ as the maximal reasoning path length.

**Case Study.** Finally, we demonstrate a few examples from both datasets to see how the retrieved reasoning path can help the PLM to make the correct prediction under the open-ended setting. As shown in Table 4, MLMs RoBERTa, DeBERTa, and RelBERT generally can only predict a single token to fill the mask. Even though they can make feasible predictions in some cases, they cannot provide valid reasoning chains. Generative language models predict the answer in an autoregressive way,

which could generate a full sentence to answer the question. However, without proper training on the test domain, even the strongest GPT-3 cannot provide a precise answer for questions like *What do people aim to do at work?*. As opposed to existing approaches, by reasoning over the external KG, KEEP can generate precise answers and provide a reasoning chain to support the answer choice without any learning steps during the inference.

## 5 Conclusion

We present an off-the-shelf framework KEEP to predict answers for open-ended commonsense reasoning without requiring answer candidates and a pre-defined answer scope. By integrating the implicit knowledge stored in PLMs and the external knowledge base, KEEP retrieves relevant reasoning paths and extracts suitable answers for commonsense questions while maintaining both efficiency and efficacy. We believe this work poses a new direction to automated commonsense reasoning under the zero-shot and open-ended setting in the Large Language Model era (Ling et al., 2023c; Zhang et al., 2023).

## Limitations and Potential Risks

Since we are one of the first to study the open-ended commonsense reasoning task, the evaluation of our model has to involve human annotators, which may bring bias to the overall evaluation. In addition, the way of prompt design may impact the language model's reasoning capability. It indicates applying this method to other tasks may require people with moderate expertise to craft a task-specific prompt to feed into the method. Lastly, we have tried multiple ways to prune and control the expansion process of the local knowledge graph, but the model scalability could be a potential issue when encountering a commonsense question with a large number of entities.

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

# A  Appendix

## A.1  Experimental Details

**Language Models.** Our method is implemented with PyTorch and the Huggingface library. We do not train any new language models but finetune existing ones with the training procedure described in Section 3.3. the language model $p_\theta$ can be any language model either with the zero-shot setting or finetuned on the external knowledge base, and we leverage the masked language model RoBERTa-large (Liu et al., 2019) since it has larger representative power in commonsense ability with a less model size (Zhou et al., 2020). Specifically for GPT-3, we used the OpenAI API and specifically chose TEXT-DAVINCI-003 as the base model.

**Knowledge Graph.** We leverage ConceptNet, a general-domain knowledge graph, as our structured knowledge source $G$, which contains 799, 273 nodes and 2, 487, 810 edges. We obtain the data from the repository[1] with version 5.6.0. Concept-Net contains 34 relations (edge types). In terms of achieving less noise and better inference time, we pre-process the ConceptNet by 1) merging similar relations into one unified relation to reduce ambiguity; 2) extracting English-only content and transforming all relations into an adjacency edge list; and 3) translating the relational edge between two concepts to a natural language by designed template: (related to, car, traffic) → "Car is related to traffic." An example of the transformation template can be found in Table 6.

**Data.** *1) CommonsenseQA (CSQA)*: Talmor et al. (2019) is a multiple-choice QA dataset about common-world scenarios, which is constructed on ConceptNet and contains 1140 test cases. *2) QASC*: Khot et al. (2020) is a multiple-choice QA dataset about grad-school science, which contains 917 test cases in total. We discard the provided answers and supporting arguments in both datasets.

---

[1]https://github.com/commonsense/conceptnet5/wiki/

| Commonsense Questions | Answer 1 | Answer 2 | Answer 3 | Answer 4 | Answer 5 | Top 1 | Top 3 | Top 5 |
|---|---|---|---|---|---|---|---|---|
| August needed money because he was afraid that he'd be kicked out of his house. What did he need money to do? | **needed for survival in urban center** | usual medium used to buy things | can buy things | gregorian calendar | root of all evil | 1 | 1 | 1 |
| The weasel was becoming a problem, it kept getting into the chicken eggs kept in the what? | found in grocery store | **hen house** | bird | roomful of junkies | farm | 0 | 1 | 1 |
| Where can you put a picture frame when it's not hung vertically? | picture frame | electro magnetic ibt | bounded surface | **table** | useful to convey idea | 0 | 0 | 1 |
| Unlike a spider and his many sight seers, people only have what? | heavier than sandwitches | go to mexican restaurants for dinner | optimistic dreams | find sound of bells mournful | watch movies at home on dvds | 0 | 0 | 0 |

Table 5: Examples of the rating criteria for assessing the model performance.

| Relation Groups | Merged Relations | Relation Text |
|---|---|---|
| antonym/distinctfrom | antonym | is the antonym of |
| atlocation/locatednear | atlocation | is at location of |
| causes/causesdesire/motivatedby | causes | causes |
| relatedto/similarto/synonym | relatedto | is related to |
| isa/instanceof/definedas | isa | is a |

Table 6: Examples of the ConceptNet edge relation transformation templates.

## A.2 More Test Cases

We illustrate more test cases of each model's performance on both datasets. (CSQA: Table 7; QASC: Table 8).

**Implementation Details.** Inferences are conducted on Nvidia Quadro RTX 6000 with approximately 100 GPU hours. We set the maximum length of the reasoning path to be 3, indicating our algorithm only searches for answers within 3-hop of neighbors from all the entities in the question. We generate $20,000$ logical sentences as the training corpus for each dataset as described in Section 3.3. We finetuned our model with 2 epochs and $1e-5$ learning rate by leveraging the training corpus.

**Human Evaluation Criteria.** We work with human annotators (students recruited from the college) to obtain the Top-N accuracy and the validity rate of the generated reasoning paths in Table 1 and Figure 6. There are three annotators in total to evaluate the generated answers. To be more specific, for each dataset, two annotators are individually assigned to score the Top-N accuracy. If there are discrepancies between two annotators' judgments, the third annotator is involved in making the final decision.

We provide guidance for evaluating each model's performance in Table 5, and the Evaluation Criteria for the results in Figure 6 are self-contained. Specifically, we sample answers five times for each model on each commonsense question and rank their answers based on their model's confidence score (i.e., Equation (2)). Human annotators are responsible for evaluating whether there exists an answer that fits the semantic meaning of the question in the Top-1, Top-3, and Top-5 candidates, respectively.

| CSQA Test Case 1 | Generation Results |
|---|---|
| | **Prompt**: Sean was in a rush to get home, but the light turned yellow and he was forced to do what? →(Sean was in a rush to get home, but the light turned yellow and he was forced to [MASK].) |
| DeBERTa-v3-large | stop |
| RoBERTa-large | something |
| RelBERT | park |
| T5-3b | He got out of his car. |
| UnifiedQA | The driver didn't turn on the lights? |
| GPT-3 | If the driver didn't turn on the lights, it could be dangerous to drive |
| KEEP (Ours) | **Stop cars**.
**Reasoning Chain**:
**Yellow light** is a type of **traffic light**, **traffic light** is capable of **stop cars**. |

| CSQA Test Case 2 | Generation Results |
|---|---|
| | **Prompt**: Reading newspaper is one of many ways to practice your what? →(Reading newspaper is one of many ways to practice your [MASK].) |
| DeBERTa-v3-large | English |
| RoBERTa-large | Writing |
| RelBERT | Writing |
| T5-3b | Skills. You'll spend plenty of time reading news items. |
| UnifiedQA | Technique. You can use print, Internet, your phone, the internet, and even more online technologies. |
| GPT-3 | Reading comprehension. |
| KEEP (Ours) | **Essential ingredient of education**.
**Reasoning Chain**:
**Newspaper** requires **Reading**, **Reading** is defined as an **Essential ingredient of education**. |

| CSQA Test Case 3 | Generation Results |
|---|---|
| | **Prompt**: A child wants to play, what would they likely want? →(A child wants to play, they would likely want [MASK].) |
| DeBERTa-v3-large | toys |
| RoBERTa-large | toys |
| RelBERT | something |
| T5-3b | Their father's toys. |
| UnifiedQA | Other children. |
| GPT-3 | A child likely would want toys, games, outdoor activities, or sports equipment. |
| KEEP (Ours) | **Toys**.
**Reasoning Chain**:
**A child** is related to **toys**. |

Table 7: Three representative test cases on the CSQA dataset.

| QASC Test Case 1 | Generation Results |
|---|---|
| | **Prompt**: Cystitis is an example of something that can be treated with? →(Cystitis is an example of something that can be treated with [MASK].) |
| DeBERTa-v3-large | surgery |
| RoBERTa-large | medicine |
| RelBERT | surgery |
| T5-3b | Illness treatment. |
| UnifiedQA | Antibiotics, but a small proportion of people with the disease are not. |
| GPT-3 | Cystitis can be treated with antibiotics, anti-inflammatory medications, and pain relievers. |
| KEEP (Ours) | **Antibiotics**. **Reasoning Chain**: **Cystitis** is a type of **Diease**, **Antibiotics** is capable of handling **Diease**. |
| QASC Test Case 2 | Generation Results |
| | **Prompt**: What allows fish to move through the water without slowing down too much? →([MASK] allows fish to move through the water without slowing down too much.) |
| DeBERTa-v3-large | It |
| RoBERTa-large | Water |
| RelBERT | Agility |
| T5-3b | In fact, they are the fastest of all living creatures on our planet. |
| UnifiedQA | Fishes all swim through the water and they all started swimming fast. |
| GPT-3 | Fish have evolved a variety of features that help them move through the water with minimal resistance. |
| KEEP (Ours) | **Fins**. **Reasoning Chain**: **Fish** is capable of **Swimming**, **Swimming** requires **Fins**. |
| QASC Test Case 3 | Generation Results |
| | **Prompt**: What made sharks excellent predators? →(Sharks are excellent predators because of [MASK].) |
| DeBERTa-v3-large | Camouflage |
| RoBERTa-large | this |
| RelBERT | something |
| T5-3b | They could not just eat their prey. |
| UnifiedQA | They have a streamlined body shape. |
| GPT-3 | Sharks have many adaptations that make them excellent predators. |
| KEEP (Ours) | **Jaws**. **Reasoning Chain**: **Sharks** is related to **Jaws**. |

Table 8: Three representative test cases on the QASC dataset.