# OpenReview forum: "Open-ended Commonsense Reasoning with Unrestricted Answer Candidates"
_EMNLP/2023/Conference — EMNLP 2023 Findings_

### Official Review · Reviewer_UBfG · 2023-08-04

**Soundness:** 4

**Excitement:**

3: Ambivalent: It has merits (e.g., it reports state-of-the-art results, the idea is nice), but there are key weaknesses (e.g., it describes incremental work), and it can significantly benefit from another round of revision. However, I won't object to accepting it if my co-reviewers champion it.

**Paper Topic And Main Contributions:**

This paper studies the problem of open-ended commonsense QA, where the answer choices or answer scopes aren't provided in the problem. The paper proposes to leverage an external knowledge graph (ConceptNet, in this work) to tackle this problem. Specifically, given a question, the approach iteratively constructs a local reasoning graph which is a sub-graph of the ConceptNet, and then extracts the best reasoning path using PLMs and cloze-style prompts. The paper experiments with the proposed approach on two datasets and compare against various zero-shot baselines, which generally shows the effectiveness of the proposed approach in the open-ended commonsense reasoning setting.


**Questions For The Authors:**

Question A: does the full KEEP approach requires labeled data for finetuning?

**Reasons To Accept:**

1) The idea of using an external knowledge graph for open-ended commonsense QA is well-motivated and effective.

2) The experiments are carefully executed, including both human evaluation and automatic evaluation of the answers. The paper also includes qualitative results in the analysis, which is helpful for understanding the pros and cons of the proposed approach.

3) The paper is clearly written and easy to understand

**Reasons To Reject:**

1) The paper could benefit from including stronger baselines.

Specifically, the paper only compares against zero-shot MLM (Roberta, Deberta) and zero-shot GLM (UnifiedQa, GPT-3) baselines. The results of the baselines are directly comparable with KEEP (without finetuning). However, IIUC, it would be unfair to compare the baselines against KEEP (with finetuning), as the full KEEP approach learns from labeled training data (line 350). I wonder if it is feasible to extend the zero-shot GLM models to finetune GLM models using labeled training data. Furthermore, as this open-ended commonsense qa setting has been relatively under-explored, providing more comprehensive baselines would be helpful for readers to gain a deeper understanding of the challenges in this setting.

2) The paper studies a very focused problem (commonsense QA without provided answer choices) which might attract a relatively narrow audience.

**Reproducibility:**

4: Could mostly reproduce the results, but there may be some variation because of sample variance or minor variations in their interpretation of the protocol or method.

**Reviewer Confidence:**

4: Quite sure. I tried to check the important points carefully. It's unlikely, though conceivable, that I missed something that should affect my ratings.

---

> ### Author Rebuttal · Authors · 2023-08-28
>
> We thank the reviewer found our paper’s idea is well-motivated and effective. Please find our detailed response as follows.
>
> *Q1. Specifically, the paper only compares against zero-shot MLM (Roberta, Deberta) and zero-shot GLM (UnifiedQa, GPT-3) baselines. The results of the baselines are directly comparable with KEEP (without finetuning). However, IIUC, it would be unfair to compare the baselines against KEEP (with finetuning), as the full KEEP approach learns from labeled training data (line 350). I wonder if it is feasible to extend the zero-shot GLM models to finetune GLM models using labeled training data.*
>
> A1. Thanks for bringing up this concern. First of all, we want to clarify the training data is only utilized to enhance PLM's ability to assess the validity of a longer reasoning path, which has little things to do with generating the correct answer. Specifically, we leverage commonsense questions and their answers to construct long-distance reasoning paths and transform them into sentences. These sentences are leveraged to finetune the PLM. We do not teach the model to directly generate the answer; instead, we want to teach the model how to reason over a knowledge graph to navigate to the most plausible answer. Given the unique focus of our finetuning process, we believe that the comparison with zero-shot models remains relevant and fair. Our finetuning serves a specific purpose distinct from direct answer generation.
>
> To further address your concern, we have conducted an additional experiment on CSQA test set. We fine-tune GLMs (T5-3B and UnifiedQA) with the same fine-tuning data and leverage LLaMA-2-70B-chat model to make the commonsense score evaluation of the best answer, please refer to the following table.
>
> |         Model         | Commonsense Score (LLaMA-2-70B-chat) |
> |:---------------------:|:------------------------------------:|
> |         T5-3B         |                 76.78                |
> |   T5-3B (Finetuned)   |                 73.25                |
> |       UnifiedQA       |                 58.92                |
> | UnifiedQA (Finetuned) |                 59.56                |
> |    KEEP (Finetuned)   |                 47.49                |
> |      Groundtruth      |                 34.67                |
>
> According to the table, providing the same finetuning data to T5-3B and UnifiedQA has little performance improvement, and zero-shot UnifiedQA can even achieve better performance than its finetuned version. The reason is KEEP’s model design considers the assessment of reasoning paths, which can inherently benefit KEEP from the provided finetuning data.
>
> *Q2: The paper studies a very focused problem (commonsense QA without provided answer choices) which might attract a relatively narrow audience.*
>
> A2: Thank you for highlighting this point. While our work specifically targets commonsense QA without providing answer choices, its implications are broader. Open-ended commonsense reasoning simulates real-world scenarios where systems must derive answers without many predefined options. This ability is crucial for AI to understand and navigate the complexities of human-like reasoning in diverse applications, from chatbots to advanced decision-support systems. Thus, while the immediate focus may seem narrow, the realistic and practical applications of our findings can benefit a wide audience across multiple AI domains.
>
> *Q3. Does the full KEEP approach requires labeled data for finetuning?*
>
> A3. We do not require labeled data to finetune the LM. As long as we can retrieve reasoning paths from knowledge graphs, and these paths can be transformed into natural language sentences, they can serve as finetuning data.

---

### Official Review · Reviewer_keAw · 2023-08-05

**Typos Grammar Style And Presentation Improvements:** Should line 393 be "without restricte…
**Soundness:** 3

**Excitement:**

4: Strong: This paper deepens the understanding of some phenomenon or lowers the barriers to an existing research direction.

**Paper Topic And Main Contributions:**

The paper studied open-ended common sense reasoning without pre-defining an answer candidate set and regulating an answer scope. The author proposed KEEP to eliminate the requirement of answer candidates by leveraging an external knowledge base, i.e., ConceptNet, as the answer 100 searching space. Besides, the proposed approach utilize the implicit knowledge stored in PLM to formulate the overall searching criteria and prune irreverent reasoning paths, i.e., and thus, reduce the searching space. Empirical results show the effectiveness of the proposed method.

**Questions For The Authors:**

The reviewer wonders whether instruction-finetuned LLMs could achieve better performance with dedicated prompts.

**Reasons To Accept:**

The paper proposed a novel approach to solve open-ended common sense reasoning without regulating an answer scope. The proposed method shows competitive results on several common sense reasoning benchmarks compared to both masked language models and generative language models. In particular,  The experiments in the paper shows the combination of Roberta-large with 355M parameters and the ConceptNet performs better than GPT-3 with 175B parameters, and thus shows the community a new way to further improve the reasoning ability of LLMs.

**Reasons To Reject:**

The proposed method solely rely on searching results in external knowledge base, and may not be applicable if the answer doesn’t exist in the knowledge base. However, sometimes the answer could be in the implicit knowledge of PLM instead of in external knowledge base. The proposed method may limit the usage of the knowledge of PLMs gained by pre-training, i.e., predict the answer using implicit knowledge in PLM.


**Reproducibility:**

4: Could mostly reproduce the results, but there may be some variation because of sample variance or minor variations in their interpretation of the protocol or method.

**Reviewer Confidence:**

3: Pretty sure, but there's a chance I missed something. Although I have a good feel for this area in general, I did not carefully check the paper's details, e.g., the math, experimental design, or novelty.

---

> ### Author Rebuttal · Authors · 2023-08-28
>
> Thanks for the constructive comments and acknowledging our work has the potential further to improve the reasoning capability of large foundation models. Please find our detailed responses to answer your questions and concerns as follows.
>
> *Q1. The proposed method relies on searching results in the external knowledge base, and may not be applicable if the answer doesn’t exist in the knowledge base. However, sometimes the answer could be in the implicit knowledge of PLM instead of in the external knowledge base. The proposed method may limit the usage of the knowledge of PLMs gained by pre-training, i.e., predict the answer using implicit knowledge in PLM.*
>
> A1. Thanks for bringing up the concern of the reliance on external knowledge bases. While our current implementation utilizes ConceptNet, our method can be coupled with other domain-specific knowledge graphs to address specialized tasks. The absence of an answer in one knowledge base doesn't restrict the model's adaptability or effectiveness, as it can easily tap into other knowledge sources as needed. In terms of the implicit knowledge of PLMs, Generative models like ChatGPT (with more than 100 billion parameters) are indeed capable of answering a broad range of questions. However, they still face challenges with domain-specific tasks due to reasons like computational overhead, hallucination, and knowledge cut-off [1]. Our method utilizes the strengths of both PLMs and external knowledge bases. While PLMs provide a broad understanding with implicit knowledge, external knowledge bases offer precision, especially in specialized domains.
>
> [1] Mialon, Grégoire, et al. "Augmented language models: a survey." arXiv preprint arXiv:2302.07842 (2023).
>
> *Q2. The reviewer wonders whether instruction-finetuned LLMs could achieve better performance with dedicated prompts.*
>
> A2. Our method is tailored to navigate the complexities of open-ended commonsense reasoning, which has proven to be a challenging task even for advanced LLMs (e.g., GPT-3). Moreover, the task of commonsense reasoning is formed with a list of questions, where each question is unique and diverse. Unlike tasks that require systematic output (e.g., information extraction and sentiment analysis) from input, there's limited commonality between commonsense questions that could be leveraged for in-context learning or supervised fine-tuning, making it a difficult task for LLMs to excel in, even with specialized prompts.
>
> *Q3. Should line 393 be "without restricted answer scope"?*
>
> A3. Sorry for the typo, we will fix this in the later version.

---

### Official Review · Reviewer_R9n2 · 2023-08-07

**Soundness:** 4

**Excitement:**

3: Ambivalent: It has merits (e.g., it reports state-of-the-art results, the idea is nice), but there are key weaknesses (e.g., it describes incremental work), and it can significantly benefit from another round of revision. However, I won't object to accepting it if my co-reviewers champion it.

**Paper Topic And Main Contributions:**

This paper tackles the problem of open-ended commonsense reasoning, which is defined as commonsense QA without a predefined set of answer options. They present a method of searching ConceptNet for the most plausible answer, and evaluate on CSQA and QASC against weak baselines.

**Reasons To Accept:**

A method of making a search path through a knowledge graph may be interesting for many problems. This paper fits today's trend of moving from constrained evaluation (e.g., classification, multiple choice) to more open-ended evaluation (instruction-following).

**Reasons To Reject:**

1. The problem that the paper scopes out is very unclear to me. Most of the paper claims to study "*open-ended commonsense reasoning*," which is defined by two characteristics: **no pre-defined answer scope** and no pre-defined answer candidates list (line 46-47). However, the method searches for and returns an answer within ConceptNet, which does seem like a predefined answer scope to me. Indeed, in the experiments, the problem is instead described as "open-ended commonsense reasoning **with restricted answer scope**," contradicting the earlier parts of the paper. This seems to be used to say there are no comparisons to prior work.
2. The overall results are very poor compared to today's models. Supervised methods have achieved ~90% accuracy on CSQA, but this paper reports its best method achieving 52.3%. Note that while answer choices are not provided, models are considered correct for providing *any* correct answer, not just the target in the original dataset. This makes the relevance of the method in the landscape of today's models unclear.
3. As mentioned before, no comparisons to prior work are included, besides UnifiedQA. Many retrieval-based methods could have been adapted to the setting with no provided answer options; or, the pretrained LMs could be allowed to retrieve over ConceptNet. For the pretrained LMs, no in-context examples are provided, and I was not able to find the prompt for left-to-right LMs in the paper.
4. The evaluation is done by humans and GPT-2. GPT-2 is a weaker model than some being evaluated (like GPT-3), and human evaluation is explained in a single sentence.

"*Since we do not have ground truth to evaluate the prediction correctness, we generate answer candidates for each commonsense question and work with human annotators to indicate whether there exists a precise answer that could answer the given question.*"

This does not describe who the annotators are (or crucial details like inter-annotator agreement) or the actual guidelines/questions they were given. It is also not stated whether they did evaluation on the entire test set, or a subset.

5. From the examples provided, the reasoning chains are invalid explanations of the answer. For instance, for the question "*What do people aim to do at work?*", the reasoning chain is "*Work is done by People, People desires to **work on new and challenging problems***" where the bolded part is the final answer. The second relation triple has nothing to do with work (as in having a job). As another example, for the question "*Cystitis is an example of something that can be treated with?*", the reasoning chain is "*Cystitis is a type of Diease, **Antibiotics** is capable of handling Diease.*" Again, the second step is completely unrelated to the disease in question, cystitis. Thus, searching through a knowledge graph seems like the entirely wrong approach to solving these problems, as can only maintain information about one entity at once when searching for the next reasoning step...

**Reproducibility:**

3: Could reproduce the results with some difficulty. The settings of parameters are underspecified or subjectively determined; the training/evaluation data are not widely available.

**Reviewer Confidence:**

3: Pretty sure, but there's a chance I missed something. Although I have a good feel for this area in general, I did not carefully check the paper's details, e.g., the math, experimental design, or novelty.

---

> ### Author Rebuttal · Authors · 2023-08-28
>
> Thanks for the review comment, please find our detailed responses as follows.
>
> *Q1. The problem that the paper scopes out is very unclear to me. The experiments are conducted as searching for answers on ConceptNet, which contradicts the definition of “unrestricted answer scope”.*
>
> A1. We understand your concern about the clarity of the problem scope. We first apologize for the typo on Line 393, we are investigating **open-ended commonsense reasoning with unrestricted answer scope**. To further address your concern, when we refer to the unrestricted answer scope, we emphasize the vastness and diversity of potential answers within our chosen knowledge source, ConceptNet. With over 300,000 nodes, each node essentially becomes a potential answer. This scale is significantly larger than typical predefined answer scopes which might consist of a limited set of conceptual entities or a set of question-related documents. Therefore, ConceptNet and other KGs don't represent a curated list of answers for specific questions.  In contrast, predefined answer scopes might be limited to a few hundred potential answers or a small subset of documents. The order of magnitude of such scopes is vastly smaller than an entire knowledge graph like ConceptNet.
>
> *Q2. The overall results are very poor compared to today's models. Supervised methods have achieved ~90% accuracy on CSQA, but this paper reports its best method achieving 52.3%.*
>
> A2. It's important to highlight the difference in task complexities. While supervised methods fine-tune pretrained LMs to select from 5 answer candidates on CSQA (20% accuracy with random guess), our approach grapples with an open-ended setting involving nearly half a million potential answer candidates (~0% accuracy with random guess). Directly comparing two settings does not provide a fair assessment, as they operate under vastly different constraints. The results we achieved are promising (better than GPT-3 in two datasets) and indicative of the potential in this direction.
>
> *Q3. As mentioned before, no comparisons to prior work are included, besides UnifiedQA. Many retrieval-based methods could have been adapted to the setting with no provided answer options; or, the pretrained LMs could be allowed to retrieve over ConceptNet.*
>
> A3. The open-ended QA problem cannot be easily solved by retrieval-based methods. Particularly, retrieval-based methods need to (pre)-train a retriever and conduct semantic search (or other similarity-based search) on external sources. However, when it comes to the commonsense reasoning problem, the retriever may not perform well. Taken a question from CSQA as an example, “A revolving door exists in hotel, where else could it be?” Without careful design, retrieval-based methods may return a list of nodes related to the “revolving door” or “hotel”, but the question asked for a place. In addition, retrieval-based methods typically operate on a pre-defined answer scope instead of a KG with million-scale nodes and relations, which further limits the adaptability in the open-ended setting.
> Our focus is on solving individual commonsense reasoning questions, which PLMs are capable of answering them directly without the need for in-context demonstrations. The prompt design is less relevant to our main contribution and thus not heavily emphasized.
>
> *Q4. The evaluation is done by humans and GPT-2. GPT-2 is a weaker model than some being evaluated (like GPT-3), and human evaluation is explained in a single sentence.*
>
> A4. Thanks for bringing up this concern, we have conducted evaluation of the *CSQA* test set based on LLaMA-2-70B-chat model to further demonstrate the performance.
>
> |       Model      | Commonsense Score (LLaMA-2-70B-chat) |
> |:----------------:|:------------------------------------:|
> | DeBERTa-v3-large |                 94.39                |
> |      RelBERT     |                 77.81                |
> |       T5-3B      |                 76.78                |
> |     UnifiedQA    |                 58.92                |
> |       GPT-3      |                 43.32                |
> | KEEP (Finetuned) |                 47.49                |
> |    Groundtruth   |                 34.67                |
>
> As can be seen from the table, the performance of each comparison method evaluated by the most advanced LLaMA-2-70B model is consistent with Table 2 (evaluated by GPT-2-large). In this table, GPT-3 is slightly better than our model since it tends to generate a coherent sentence rather than a single word, which may induce a better commonsense score.
>
> Due to space limit, the Human Evaluation Criteria is summarized in Appendix. We will include more details in the revised version if the manuscript can be accepted.
>
> Q5. The reasoning chains are invalid explanations of the answer. For instance, for the question "What do people aim to do at work?", the reasoning chain is "Work is done by People, People desires to work on new and challenging problems" where the bolded part is the final answer. The second relation triple has nothing to do with work (as in having a job). … Thus, searching through a KG seems like the entirely wrong approach to solving these problems, as can only maintain information about one entity at once when searching for the next reasoning step.
>
> *A5. The comment seems to be based on the assumption that all entities in the reasoning chain have to be related to the initial entity. However, based on the definition of reasoning chain [1, 2], multi-hop reasoning allows the agent reach one or more intermediate conclusions before concluding the final answer and each of the intermediate conclusions serves as a necessary premise for the next one. This sequence of intermediate and final conclusions is called a reasoning chain, and the conclusion does not have to be directly related to the initial entity. Retrieving reasoning chain to solve multi-hop reasoning task is a typical approach in recent literature [3].*
>
> When searching for the next reasoning step, our algorithm also considers previous retrieved relation triplets. As described in Section 3.3 (Line 361-375) and Equation (2), our method retrieves all relevant reasoning paths from the KG and obtains various answer candidates. The final answer is chosen based on an aggregated commonsense score so it considers **all retrieved reasoning triplets**.
>
> [1] Jhamtani, Harsh, and Peter Clark. "Learning to Explain: Datasets and Models for Identifying Valid Reasoning Chains in Multihop Question-Answering." EMNLP, 2020.
> [2] Xu, Weiwen, et al. "Exploiting Reasoning Chains for Multi-hop Science Question Answering." Findings of EMNLP. 2021.
> [3] Lin, Xi Victoria, et al. "Multi-Hop Knowledge Graph Reasoning with Reward Shaping." EMNLP. 2018.

---

### Meta-Review · Area_Chair_etKj · 2023-09-20

**Recommendation:** 3

**Metareview:**

The authors propose a framework called KEEP, designed for predicting answers in open-ended commonsense reasoning tasks. KEEP extracts reasoning paths from an external knowledge base via concept extraction, entity linking, and knowledge graph expansion. The method incorporates the implicit knowledge in the PLM to identify relevant reasoning paths (pruning irrelevant ones). The relevant reasoning paths are used either in a zero-shot setting or as training instances to finetune the PLM.

The paper introduces a novel approach to address open-ended commonsense reasoning without restricting the answer scope. The paper reports competitive performance on two commonsense reasoning benchmarks (CSQA and QASC) when compared to both masked and generative language models.

Nonetheless, the empirical validation on the CSQA benchmark presents a major limitation. The proposed approach extracts reasoning paths from ConceptNet to solve CSQA, although the CSQA benchmark was created based on the ConceptNet knowledge base itself (Note that the CSQA leaderboard no longer accepts submissions that make use of ConceptNet). The reviewers have also suggested improvements to strengthen the paper's contribution, including clarifying the problem definition and conducting comparisons with stronger baselines.

---

### Decision · Program_Chairs · 2023-10-07

**Decision:**

Accept-Findings

**Comment:**

The authors propose a framework called KEEP, designed for predicting answers in open-ended commonsense reasoning tasks. KEEP extracts reasoning paths from an external knowledge base via concept extraction, entity linking, and knowledge graph expansion. The method incorporates the implicit knowledge in the PLM to identify relevant reasoning paths (pruning irrelevant ones). The relevant reasoning paths are used either in a zero-shot setting or as training instances to finetune the PLM.

The paper introduces a novel approach to address open-ended commonsense reasoning without restricting the answer scope. The paper reports competitive performance on two commonsense reasoning benchmarks (CSQA and QASC) when compared to both masked and generative language models.

Nonetheless, the empirical validation on the CSQA benchmark presents a major limitation. The proposed approach extracts reasoning paths from ConceptNet to solve CSQA, although the CSQA benchmark was created based on the ConceptNet knowledge base itself (Note that the CSQA leaderboard no longer accepts submissions that make use of ConceptNet). The reviewers have also suggested improvements to strengthen the paper's contribution, including clarifying the problem definition and conducting comparisons with stronger baselines.